# Winning Tickets from Random Initialization: Aligning Masks for Sparse Training

**Rohan Jain**[*1], **Mohammed Adnan**[*1,3], **Ekansh Sharma**[2,3], **Yani Ioannou**[1]
[1]University of Calgary  [2]University of Toronto  [3]Vector Institute for AI
{rohan.jain1,adnan.ahmad,yani.ioannou}@ucalgary.ca, ekansh@cs.toronto.edu

## Abstract

The Lottery Ticket Hypothesis (LTH) suggests that there exists a sparse *winning ticket* mask and weights that achieves the same generalization performance as the dense model while using much fewer parameters. LTH achieves this by iteratively sparsifying and re-training within the pruned solution basin. This procedure is expensive, and a winning ticket's sparsity mask does not generalize to other weight initializations. Recent work has suggested that Deep Neural Networks (DNNs) trained from random initialization find solutions within the same basin modulo weight symmetry, and proposed a method to align trained models within the same basins. We propose permuting the winning ticket mask to align with the new optimization basin when performing sparse training from a different random initialization than the one used to derive the pruned mask. Using this permuted mask, we show it is possible to significantly increase the generalization performance of sparse training from random initialization with the permuted mask as compared to sparse training naively using the non-permuted mask.

## 1   Introduction

In recent years, foundation models have achieved state-of-the-art results for different tasks. However, such an exponential increase in the size of state-of-the-art models requires a similarly exponential increase in the memory and computational costs required to train, store and use these models — decreasing the accessibility of these models for researchers and practitioners alike. Seminal works have demonstrated that large models can be pruned after training with minimal loss in accuracy [13, 17]. While model pruning makes inference more efficient, it does not reduce the computational cost of training the model. Motivated by training a sparse model from a random initialization, the Lottery Ticket Hypothesis (LTH) proposes to solve the sparse training problem by reusing the same initialization as used to train the pruned models. On very small models, training from such an initialization maintains the generalization performance of the pruned model and demonstrates that training with a highly sparse mask is possible [10]. In practice however, when training even modestly-sized models, *weight rewinding* [11] is necessary — requiring significantly more compute than dense training alone.

Our hypothesis is that in order to reuse the LTH mask for different random initialization, the winning ticket mask obtained from LTH needs to be permuted such that it aligns with the optimization basin associated with this new initialization. We illustrate this intuition in Fig. 1. To empirically validate our hypothesis, we obtain a sparse mask using iterative magnitude pruning (IMP) on model $A$ (from Fig. 1) and show that given a permutation that aligns the optimization basin of model $A$ and a new random initialization, the mask can be reused. The sparse model (with the permuted mask) can be trained to match the generalization performance of the LTH solution.

---

[*]Equal contribution.

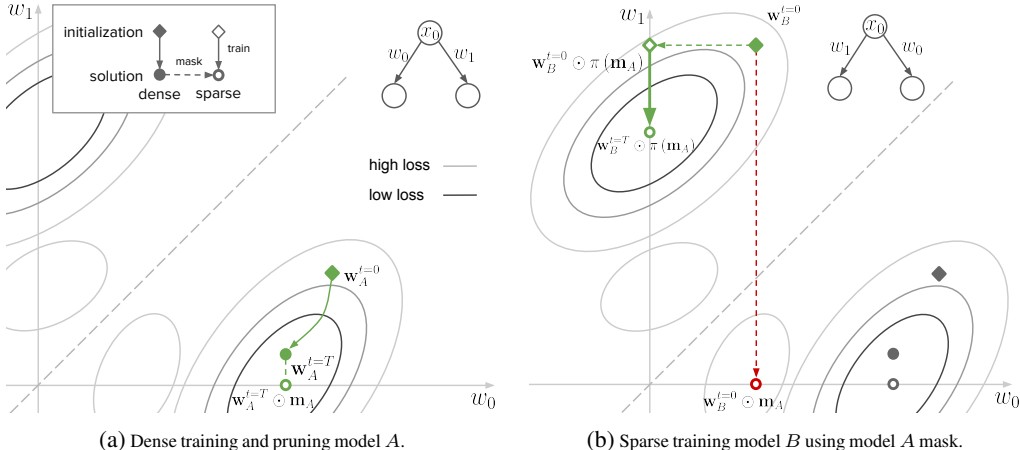

(a) Dense training and pruning model $A$.    (b) Sparse training model $B$ using model $A$ mask.

Figure 1: **Weight Symmetry and the Sparse Training Problem**. A model with a single layer and only two parameters, $\mathbf{w} = (w_0, w_1)$, operating on a single input scale $x_0$ has the weight symmetry in the 2D loss landscape as illustrated above. In (a) the original dense model, $\mathbf{w}_A$, is trained from a random dense initialization, $\mathbf{w}_A^{t=0}$ to a dense solution, $\mathbf{w}_A^{t=T}$, which is then pruned using weight magnitude resulting in the mask $\mathbf{m}_A = (1,0)$. In (b), naively using the same mask to train a model, B, from a different random initialization will likely result in the initialization being far from a good solution. Permuting the mask to match the (symmetric) basin in which the new initialization is in will enable sparse training.

## 2   Background & Related Work

**Weight Symmetry.**    The process of training DNN's requires optimizing a non-linear function over a non-convex loss landscape consisting of numerous local minima, narrow ravines, plateaus, saddle points and *basins*. [4, 5, 7, 15, 26, 28, 37]. Despite non-convex optimization problems being NP-hard [31], the nature of first order stochastic optimizers such as SGD [30] have been theoretically proven to be highly effective in optimizing DNN's [18, 21] and in practice as well. Empirical evidence suggests that when training independent NN's using SGD, with different batch orders and initializations, the resulting training trajectories often exhibit remarkable similarities [1, 36]. With the rise of building larger NN's, this puzzling phenomenon has been attributed to overparameterization, which is responsible for creating numerous minima in the loss landscape, resulting in multiple different functions which fit the data similarly [23, 27]. However, as early as the 1990s, Hecht-Nielsen [20] demonstrated that neural networks are *permutation invariant* possessing a weight-symmetrical property, where swapping any two neurons within a hidden layer does not alter the underlying function being computed. In other words, the permuted network remains functionally equivalent to its original configuration. Hence, the existence of permutation symmetries in the loss landscape contribute to its non-convexity, as it creates copies of global minima at different points in weight space [3, 8, 14].

**Linear Mode Connectivity *modulo* Permutation.**    Typically, linearly interpolating between the weights of two independently trained networks usually results in a higher loss/0–1 error compared to the two endpoints. Entezari et al. [8] conjectured, that independently obtained SGD solutions have no error barrier if one accounts for the permutation symmetries. Building on this conjecture, several algorithms have been developed to address permutation invariance by aligning trained networks to the same optimization basin [1, 22, 34, 35]. Ainsworth et al. [1] demonstrate that DNN's trained from random initialization find solutions within the same basin modulo permutation symmetry. They proposed three algorithms to permute the units of one model to align it with a reference model, enabling the permuted model to exhibit LMC (i.e. reduced loss barrier) with the reference model. Benzing et al. [2] use a permutation found after training to exhibit LMC between networks at initialization. The use of activation matching for model alignment was originally introduced by Li et al. [25] to ensure models learn similar representations when performing the same task. A rigorous study from Sharma et al. [32] introduced a notion of *simultaneous weak linear connectivity* where a permutation, $\pi$ aligning two networks also simultaneously aligns two larger fully trained networks throughout the entire SGD trajectory and the same $\pi$ also aligns successive iterations of independently sparsified networks found via IMP. This work also provided early evidence towards re-usability of sparse masks.

**Lottery Ticket Hypothesis.** Neural network pruning is a highly effective method of reducing the parameters in a trained dense neural network, pruning as many as 85–95% of weights while not significantly affecting generalization performance [16, 17]. The LTH proposes to solve the sparse training problem by re-using the same initialization as used to train the pruned models. For very small models, training from such an initialization maintains the generalization performance of the pruned model, and demonstrates that training with a highly sparse mask is possible [10]. In practice however, subsequent work has shown that when training modestly-sized models requires using *weight rewinding* [11] — requiring significantly more compute than dense training alone. Furthermore, recent work has shown that the LTH effectively re-learns the pruned solution [9]. To make any practical use of sparse training, finding methods of sparse training from random initialization may be necessary to both find new solutions, and realize any efficiency gains in training.

## 3 Method

**Motivation.** In this work, we try to understand *why LTH masks fail to transfer to a new random initialization*. Our hypothesis is that the loss basin corresponding to the LTH mask is not aligned with the new random initialization as shown in Fig. 1. Since the sparse mask is not in alignment with the basin for the new random initialization, sparse training does not work well; therefore, aligning the LTH mask with new random initialization will improve sparse training and enable the transfer of LTH masks to random initializations.

**Aligning Masks via Weight Symmetry.** Ainsworth et al. [1] showed the permutation symmetries of the weight space can be leveraged to align the basin of two models trained from different random initializations. In their approach, the authors utilize activation matching to align the activations of two models. By permuting the parameters of the second model, they maximize the correlation between the activations of the first and second models. This method fits within the framework of solving a linear assignment problem (LAP), enabling efficient computation. In our experiments, we train two dense models, $\mathbf{w}_A^{t=0}$ and $\mathbf{w}_B^{t=0}$, to convergence and then use activation matching (implemented by Jordan et al. [22]) to find the permutation mapping $\pi$, such that the activations of $\pi(\mathbf{w}_A^{t=T})$ and $\mathbf{w}_B^{t=T}$ are aligned. Mask $\mathbf{m}_A$, obtained using IMP is also permuted with the same permutation map $\pi$. The intuition is that the permuted mask is aligned with the loss basin of model $\mathbf{w}_B^{t=T}$ and thus can be optimized easily (refer to Fig. 2). We denote training with the permuted mask, $\pi(\mathbf{m}_A)$ as *permuted* and with the non-permuted mask, $\mathbf{m}_A$ as *naive*. More details in Appendix A.5.

**Sparse Training.** For evaluating the transferability of LTH masks, we use a new random initialization $\mathbf{w}_B^{t=0}$ and sparse masks $\mathbf{m}_A$ and $\pi(\mathbf{m}_A)$ for sparse training the naive and permuted solution respectfully. We also evaluate the LTH baseline, i.e., training model $\mathbf{w}_A^{t=0}$ with mask $\mathbf{m}_A$. Since LTH requires weight rewinding to an earlier point in training, we also use a rewound checkpoint from epoch $t = k \ll T$ for both the baselines and permuted solution.

## 4 Results

To validate our hypothesis, we trained ResNet20 [19] and VGG11 [33] models on CIFAR-10/100 datasets [24] (details in Appendix A.1). We used the same set of hyper-parameters for training different sparse baselines and our permuted solution (details in Appendix A.1).

**ResNet.** We trained ResNet20 on CIFAR-10/100 datasets. As shown in Figs. 4 and 5, consistent across varying width and different datasets both LTH and permuted solution improve as the rewind point increases in contrast to the naive solution, which does not improve on increasing the rewind point. We observed that naive performance saturates after $k \geq 50$ and does not yield further improvement. Since it is more difficult to train models with higher sparsity, the gap between naive and permuted solutions increases as sparsity increases, as shown in Figs. 4d, 4h and 4l. The improved performance of permuted solution supports our hypothesis and shows that misalignment of LTH masks and loss basin corresponding to new random initialization may be the reason why LTH masks do not transfer to different initializations. We also show accuracy vs sparsity plots for $k = \{10, 25, 50, 100\}$ (details in Appendix A.4); as sparsity increases, the gap between permuted and naive solution increases for all rewind points (see Appendix A.3). Results for CIFAR-100 dataset is shown in Fig. 6 in Appendix A.3.

**VGG11.** We utilize the modified VGG11 architecture implemented by Jordan et al. [22] trained on CIFAR-10 (details in Appendix A.1). We observe that for a moderate sparsity Fig. 3a, the permuted and naive solution are relatively similar and steadily increasing together as we increase the rewind

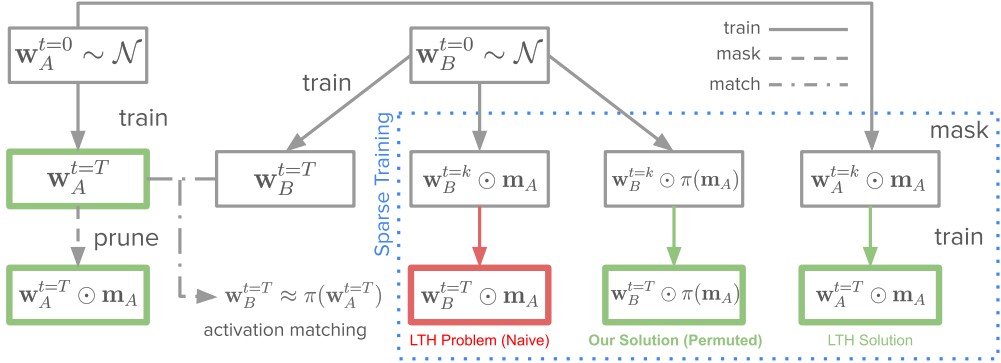

Figure 2: The overall framework of the training procedure, beginning with two distinct dense random weight initializations, $\mathbf{w}_A^{t=0}$, $\mathbf{w}_B^{t=0}$ sampled from a normal distribution, $\mathcal{N}$. The sparse training problem attempts to train the random initialization, $\mathbf{w}_B^{t=0}$ using the naive mask $\mathbf{m}_A$, found by pruning a dense trained model, $\mathbf{w}_A^{t=T}$. However, this results in poor generalization performance [12]. We propose to instead train $\mathbf{w}_B^{t=k}$ at some rewound epoch $k$, equipped with a *permuted* mask $\pi(\mathbf{m}_A)$. We show that this achieves more comparable generalization to the pruned model/trained LTH solution, $\mathbf{w}_A^{t=T} \odot \mathbf{m}_A$. More details in Appendix A.5.

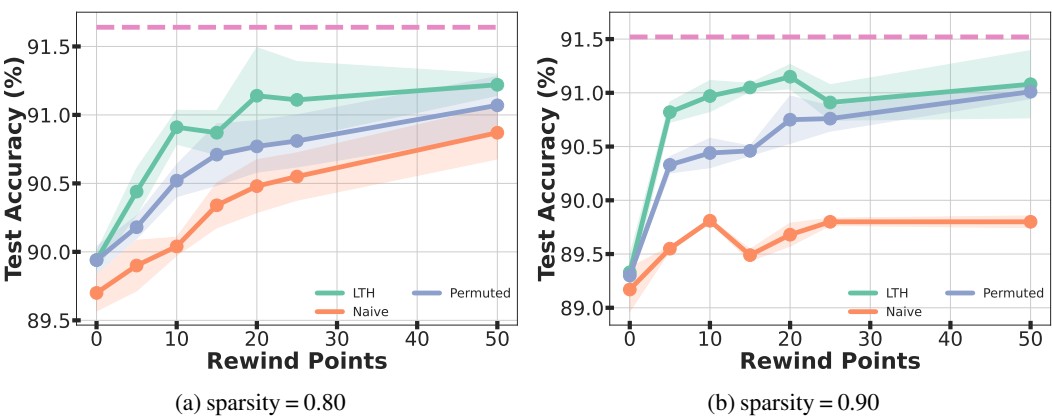

(a) sparsity = 0.80          (b) sparsity = 0.90

Figure 3: **VGG11$\times${1}/CIFAR-10.** Test accuracy of sparse network solutions at increasing rewound points at various sparsity levels. The dashed (- -) line shows the dense model accuracy. **Note**, we do not include the plots for sparsity = {0.95, 0.97}, because the naive solution performs poorly yielding consistent metrics of test accuracy = 10% and test loss $\approx \log_e(10)$ throughout all rewind points.

points with the permuted solution consistently taking the slight edge over naive. As sparsity increases in Fig. 3b, a significant gap begins to emerge between the permuted and naive solutions. As the rewind point increases, the permuted solution gradually improves and approaches the performance of LTH, while the naive solution significantly plateaus for $k \geq 20$ and performance subsides.

**Effect of width multiplier.** The activation matching algorithm proposed by Ainsworth et al. [1] does not find the global optimum; rather, it uses a greedy search to explore a restricted solution space. The resulting permutation mapping aligns well in practice, especially for wider models [1]. We also examine how increasing width multipliers affect aligning ResNets in weight space. As illustrated in Fig. 4, the permuted solutions increasingly match the LTH solution as the model width expands. This finding is notable, as Ainsworth et al. [1] also observed improved weight matching in wider models, an effect further validated by Sharma et al. [32] for activation matching. This trend may also help explain why the VGG results in Fig. 3—an exceptionally over-parameterized model for CIFAR-10—are so close to the LTH baseline. If the hypothesis by Ainsworth et al. [1] holds—neural network loss landscapes nearly contain a single solution basin modulo weight symmetry—then with an ideal permutation mapping, the permuted solution would match the LTH solution. However, our experiments still seem to corroborate our hypothesis and may provide insights into why LTH does not transfer well to new initializations.

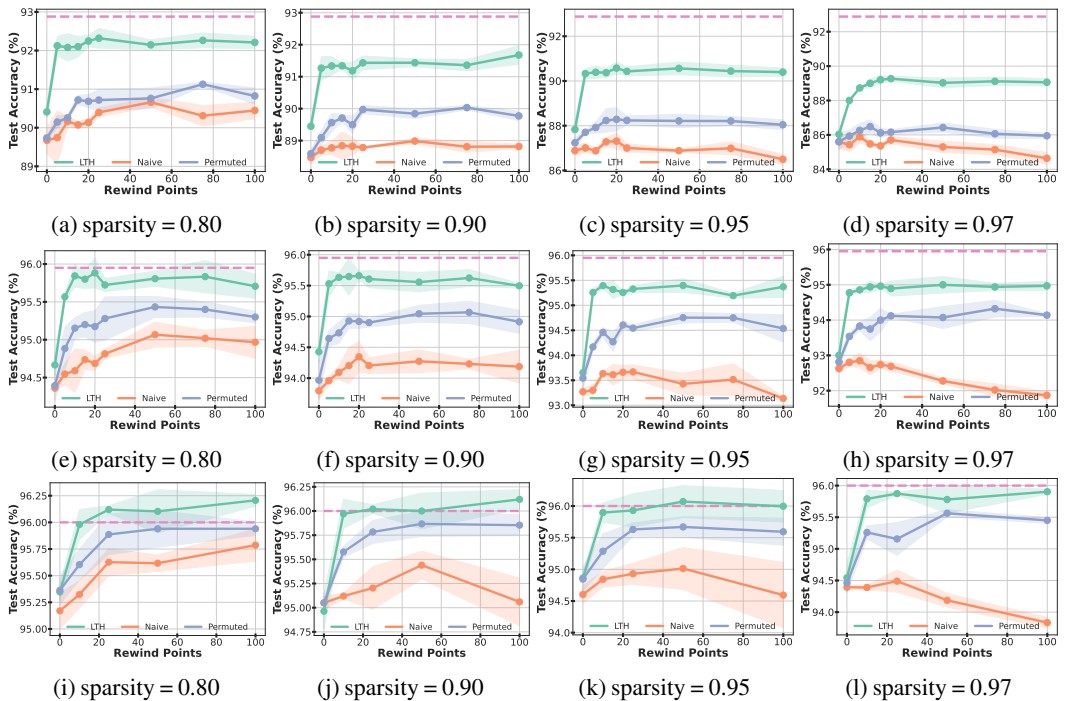

Figure 4: **ResNet20×{1,4,8}/CIFAR-10**. The top, middle, and bottom rows correspond to widths of 1, 4, and 8, respectively. The effect of the rewind points on the test accuracy for different sparsities is shown. As the sparsity and rewind epoch increase, the gap between training from a random initialization with the permuted mask and the LTH/dense baseline (dashed line) decreases, unlike training with a non-permuted mask (naive).

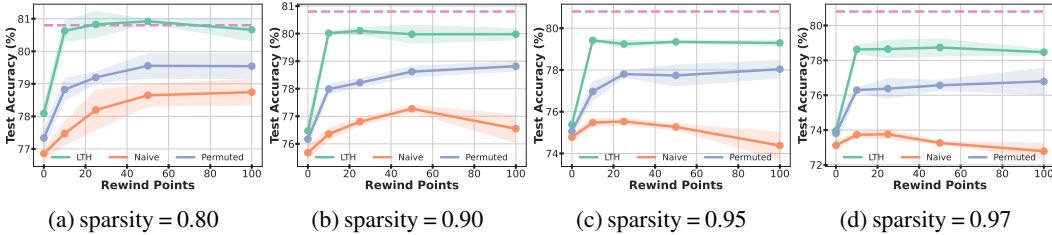

Figure 5: **ResNet20×{8}/CIFAR-100**. Trained ResNet20 with a width of 8 on CIFAR-100. Our results show that, as sparsity increases, the gap between the permuted and naive solutions increase and the permuted solution gradually approaches the LTH baseline. The dashed (- -) line shows the dense model accuracy.

# 5 Conclusion

Sparse training and the Lottery Ticket Hypothesis (LTH) have gained significant traction in recent years. In this work, we seek to deepen insights into sparse training from random initialization and the LTH by leveraging permutation invariance in DNNs. Our empirical findings across various models and datasets support the hypothesis that misalignment between the mask and loss basin prevents effective use of LTH masks with new initializations. One limitation is that activation matching is weaker for narrow models; future work will explore more efficient matching algorithms.

# 6 Acknowledgements

We acknowledge the support of Alberta Innovates (ALLRP-577350-22, ALLRP-222301502), the Natural Sciences and Engineering Research Council of Canada (RGPIN-2022-03120, DGECR-2022-00358), and Defence Research and Development Canada (DGDND-2022-03120). This research was enabled in part by support provided by the Digital Research Alliance of Canada (alliancecan.ca).

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

# A Appendix

## A.1 Implementation

**Architectures** For residual neural networks, we train the standard ResNet20 on CIFAR-10 and CIFAR-100 with varying width. We implemented a scalar, $w$, that adjusts the number of channels in each convolutional and fully connected layer:

- **First Convolution Layer**: The number of output channels is scaled from 16 to $w \times 16$.
- **Layer 1,2,3**: The number of output channels for the convolutional blocks in these layers are scaled from 16, 32, and 64 to $w \times 16$, $w \times 32$, and $w \times 64$, respectively.
- **Fully Connected Layer**: The input dimension to the final linear layer is scaled to $w \times 64$.

For convolutional neural networks, we train a modified version of the standard VGG11 implemented by [22] on CIFAR-10. Primary differences are:

- A single fully connected layer at the end which directly maps the flattened feature map output from the convolutional layers to the 10 classes for CIFAR-10 classification.
- The classifier is set up for CIFAR-10 with 10 output classes as originally VGG11 was designed for ImageNet with 1000 output classes [6].

Each of our results for a given rewound point, $k$, is averaged over 3 runs with distinct seeds.

**Datasets** For our set of experiments we used the CIFAR-10 and CIFAR-100 datasets [24]. We apply the following standard data augmentation techniques to the training set:

- `RandomHorizontalFlip`: Randomly flips the image horizontally with a given probability (by default, 50%).
- `RandomCrop`: Randomly crops the image to a size of $32 \times 32$ pixels, with a padding of 4 pixels around the image.

**Optimizers** We use the same hyperparameter settings for both architectures on both datasets outlined in Table 1.

| Hyperparameter | Value |
|---|---|
| Optimizer | SGD |
| Momentum | 0.9 |
| Dense Learning Rate | 0.08 |
| Sparse Learning Rate | 0.02 |
| Weight Decay | $5 \times 10^{-4}$ |
| Batch Size | 128 |

Table 1: Hyperparameters used for dense and sparse training.

**Pruning** We apply standard IMP [10, 16] *without weight rewinding* to obtain our final mask, $\mathbf{m}_A$, producing a sparse subnetwork $\mathbf{w}_A^{t=T} \odot \mathbf{m}_A$. For pruning, we utilize PyTorch's `torch.nn.utils.prune` library [29].

1. In an unstructured, global manner, we identify and mask (set to zero) the smallest 20% of unpruned weights based on their magnitude.
2. This process is repeated for $s$ rounds to achieve the target sparsity $S$, with each subsequent round pruning 20% of the remaining weights.
3. During each round, the model is trained for `train_epochs_per_prune` epochs with the intermediate mask applied.

The hyperparameters in Table 2 is used for pruning the ResNet20 and VGG11 architectures.

| Hyperparameter | Value |
|---|---|
| `train_epochs_per_prune` | 50 |
| Learning Rate | 0.01 |

Table 2: Hyperparameters used for pruning.

Table 3: **ResNet20×{1}/CIFAR-10**. Results using the ResNet20×{1} trained on CIFAR-10, from a rewind point $k$, using various methods of sparse training with sparsity $S$. LTH trains within the original dense/pruned solution basin, while naive/permuted train from a new random initialization.

| $S$ | Method | Rewind Epoch $k$ | | | | | | | | |
|---|---|---|---|---|---|---|---|---|---|---|
| | | $k=0$ | 5 | 10 | 15 | 20 | 25 | 50 | 75 | 100 |
| 80% | LTH | $90.41 \pm 0.14$ | $92.12 \pm 0.25$ | $92.08 \pm 0.36$ | $92.10 \pm 0.27$ | $92.25 \pm 0.14$ | $92.32 \pm 0.26$ | $92.15 \pm 0.13$ | $92.26 \pm 0.19$ | $92.21 \pm 0.16$ |
| | naive | $89.67 \pm 0.35$ | $89.74 \pm 0.69$ | $90.16 \pm 0.14$ | $90.07 \pm 0.09$ | $90.13 \pm 0.11$ | $90.40 \pm 0.11$ | $90.66 \pm 0.12$ | $90.31 \pm 0.27$ | $90.45 \pm 0.22$ |
| | perm. | $89.74 \pm 0.05$ | $90.15 \pm 0.16$ | $90.26 \pm 0.08$ | $90.72 \pm 0.12$ | $90.68 \pm 0.18$ | $90.72 \pm 0.28$ | $90.76 \pm 0.27$ | $\mathbf{91.13 \pm 0.06}$ | $90.82 \pm 0.21$ |
| 90% | LTH | $89.45 \pm 0.10$ | $91.27 \pm 0.37$ | $91.34 \pm 0.28$ | $91.34 \pm 0.29$ | $91.18 \pm 0.27$ | $91.43 \pm 0.22$ | $91.44 \pm 0.12$ | $91.36 \pm 0.18$ | $91.68 \pm 0.28$ |
| | naive | $88.47 \pm 0.21$ | $88.70 \pm 0.14$ | $88.77 \pm 0.21$ | $88.84 \pm 0.43$ | $88.83 \pm 0.27$ | $88.78 \pm 0.02$ | $88.99 \pm 0.08$ | $88.81 \pm 0.17$ | $88.82 \pm 0.07$ |
| | perm. | $88.59 \pm 0.11$ | $89.09 \pm 0.22$ | $89.56 \pm 0.28$ | $89.71 \pm 0.12$ | $89.50 \pm 0.27$ | $89.97 \pm 0.13$ | $89.84 \pm 0.15$ | $\mathbf{90.03 \pm 0.07}$ | $89.77 \pm 0.15$ |
| 95% | LTH | $87.83 \pm 0.38$ | $90.33 \pm 0.22$ | $90.39 \pm 0.28$ | $90.37 \pm 0.21$ | $90.58 \pm 0.26$ | $90.43 \pm 0.20$ | $90.56 \pm 0.29$ | $90.44 \pm 0.26$ | $90.40 \pm 0.19$ |
| | naive | $86.89 \pm 0.21$ | $87.01 \pm 0.23$ | $86.88 \pm 0.13$ | $87.28 \pm 0.19$ | $87.31 \pm 0.36$ | $87.00 \pm 0.19$ | $86.88 \pm 0.08$ | $86.99 \pm 0.29$ | $86.50 \pm 0.22$ |
| | perm. | $87.24 \pm 0.22$ | $87.70 \pm 0.08$ | $87.92 \pm 0.25$ | $88.23 \pm 0.52$ | $\mathbf{88.29 \pm 0.52}$ | $\mathbf{88.24 \pm 0.20}$ | $88.21 \pm 0.30$ | $88.21 \pm 0.20$ | $88.04 \pm 0.22$ |
| 97% | LTH | $86.03 \pm 0.22$ | $88.00 \pm 0.02$ | $88.73 \pm 0.05$ | $89.00 \pm 0.24$ | $89.21 \pm 0.23$ | $89.27 \pm 0.14$ | $89.03 \pm 0.27$ | $89.12 \pm 0.25$ | $89.06 \pm 0.21$ |
| | naive | $85.60 \pm 0.38$ | $85.43 \pm 0.40$ | $85.89 \pm 0.37$ | $85.48 \pm 0.13$ | $85.36 \pm 0.14$ | $85.70 \pm 0.21$ | $85.30 \pm 0.32$ | $85.14 \pm 0.29$ | $84.64 \pm 0.34$ |
| | perm. | $85.61 \pm 0.48$ | $85.93 \pm 0.34$ | $86.26 \pm 0.40$ | $\mathbf{86.48 \pm 0.39}$ | $86.12 \pm 0.27$ | $86.16 \pm 0.14$ | $86.43 \pm 0.27$ | $86.06 \pm 0.26$ | $85.95 \pm 0.14$ |

Table 4: **ResNet20×{4}/CIFAR-10**. Same as Table 3 except using a width-multiplier of 4.

| $S$ | Method | Rewind Epoch $k$ | | | | | | | | |
|---|---|---|---|---|---|---|---|---|---|---|
| | | $k=0$ | 5 | 10 | 15 | 20 | 25 | 50 | 75 | 100 |
| 80% | LTH | $94.67 \pm 0.14$ | $95.57 \pm 0.05$ | $95.84 \pm 0.15$ | $95.80 \pm 0.12$ | $95.88 \pm 0.20$ | $95.72 \pm 0.09$ | $95.81 \pm 0.10$ | $95.83 \pm 0.21$ | $95.71 \pm 0.16$ |
| | naive | $94.36 \pm 0.04$ | $94.55 \pm 0.14$ | $94.59 \pm 0.29$ | $94.74 \pm 0.13$ | $94.69 \pm 0.09$ | $94.81 \pm 0.06$ | $95.07 \pm 0.17$ | $95.02 \pm 0.11$ | $94.97 \pm 0.21$ |
| | perm. | $94.39 \pm 0.19$ | $94.88 \pm 0.28$ | $95.15 \pm 0.14$ | $95.20 \pm 0.16$ | $95.17 \pm 0.21$ | $95.28 \pm 0.29$ | $\mathbf{95.43 \pm 0.14}$ | $\mathbf{95.40 \pm 0.10}$ | $95.30 \pm 0.08$ |
| 90% | LTH | $94.43 \pm 0.17$ | $95.53 \pm 0.21$ | $95.63 \pm 0.07$ | $95.65 \pm 0.30$ | $95.66 \pm 0.07$ | $95.61 \pm 0.14$ | $95.56 \pm 0.16$ | $95.62 \pm 0.14$ | $95.50 \pm 0.04$ |
| | naive | $93.79 \pm 0.15$ | $93.96 \pm 0.05$ | $94.09 \pm 0.11$ | $94.20 \pm 0.29$ | $94.35 \pm 0.25$ | $94.20 \pm 0.13$ | $94.27 \pm 0.19$ | $94.23 \pm 0.08$ | $94.19 \pm 0.27$ |
| | perm. | $93.97 \pm 0.29$ | $94.64 \pm 0.13$ | $94.73 \pm 0.17$ | $94.93 \pm 0.12$ | $94.92 \pm 0.11$ | $94.90 \pm 0.07$ | $\mathbf{95.04 \pm 0.14}$ | $\mathbf{95.07 \pm 0.18}$ | $94.91 \pm 0.19$ |
| 95% | LTH | $93.65 \pm 0.12$ | $95.26 \pm 0.08$ | $95.39 \pm 0.05$ | $95.32 \pm 0.18$ | $95.26 \pm 0.03$ | $95.33 \pm 0.07$ | $95.40 \pm 0.14$ | $95.19 \pm 0.05$ | $95.37 \pm 0.21$ |
| | naive | $93.27 \pm 0.07$ | $93.30 \pm 0.11$ | $93.63 \pm 0.04$ | $93.61 \pm 0.21$ | $93.66 \pm 0.13$ | $93.67 \pm 0.14$ | $93.43 \pm 0.21$ | $93.51 \pm 0.32$ | $93.14 \pm 0.03$ |
| | perm. | $93.54 \pm 0.24$ | $94.17 \pm 0.07$ | $94.46 \pm 0.10$ | $94.27 \pm 0.19$ | $94.61 \pm 0.07$ | $94.54 \pm 0.07$ | $\mathbf{94.75 \pm 0.11}$ | $\mathbf{94.75 \pm 0.09}$ | $94.54 \pm 0.27$ |
| 97% | LTH | $93.00 \pm 0.11$ | $94.77 \pm 0.09$ | $94.86 \pm 0.06$ | $94.94 \pm 0.17$ | $94.96 \pm 0.06$ | $94.89 \pm 0.21$ | $95.00 \pm 0.24$ | $94.94 \pm 0.10$ | $94.97 \pm 0.13$ |
| | naive | $92.63 \pm 0.12$ | $92.80 \pm 0.10$ | $92.85 \pm 0.21$ | $92.66 \pm 0.21$ | $92.74 \pm 0.11$ | $92.69 \pm 0.14$ | $92.28 \pm 0.09$ | $92.02 \pm 0.18$ | $91.87 \pm 0.10$ |
| | perm. | $92.81 \pm 0.27$ | $93.54 \pm 0.08$ | $93.83 \pm 0.12$ | $93.75 \pm 0.34$ | $94.00 \pm 0.33$ | $94.12 \pm 0.04$ | $94.07 \pm 0.31$ | $\mathbf{94.32 \pm 0.24}$ | $94.14 \pm 0.04$ |

Table 5: **VGG11×{1}/CIFAR-10**. Results using the VGG11, trained on CIFAR-10, from a rewind point $k$, using various methods of sparse training with sparsity $S$.

| $S$ | Method | Rewind Epoch $k$ | | | | | | |
|---|---|---|---|---|---|---|---|---|
| | | $k=0$ | 5 | 10 | 15 | 20 | 25 | 50 |
| 80% | LTH | $89.94 \pm 0.06$ | $90.44 \pm 0.17$ | $90.91 \pm 0.12$ | $90.87 \pm 0.16$ | $91.14 \pm 0.28$ | $91.11 \pm 0.08$ | $91.22 \pm 0.08$ |
| | naive | $89.70 \pm 0.13$ | $89.90 \pm 0.18$ | $90.04 \pm 0.07$ | $90.34 \pm 0.16$ | $90.48 \pm 0.19$ | $90.55 \pm 0.17$ | $90.87 \pm 0.19$ |
| | perm. | $89.94 \pm 0.1$ | $90.18 \pm 0.08$ | $90.52 \pm 0.17$ | $90.71 \pm 0.22$ | $90.77 \pm 0.19$ | $90.81 \pm 0.19$ | $\mathbf{91.07 \pm 0.21}$ |
| 90% | LTH | $89.33 \pm 0.16$ | $90.82 \pm 0.09$ | $90.97 \pm 0.14$ | $91.05 \pm 0.04$ | $91.15 \pm 0.11$ | $90.91 \pm 0.17$ | $91.08 \pm 0.31$ |
| | naive | $89.17 \pm 0.2$ | $89.55 \pm 0.02$ | $89.81 \pm 0.02$ | $89.49 \pm 0.05$ | $89.68 \pm 0.11$ | $89.80 \pm 0.03$ | $89.80 \pm 0.05$ |
| | perm. | $89.30 \pm 0.02$ | $90.33 \pm 0.08$ | $90.44 \pm 0.14$ | $90.46 \pm 0.04$ | $90.75 \pm 0.22$ | $90.76 \pm 0.12$ | $\mathbf{91.01 \pm 0.06}$ |

## A.2 Results

Detailed results for ResNet20×{1}, ResNet20×{4}, and VGG11×{1} experiments are provided in Table 3, Table 4, and Table 5 respectively.

## A.3 Additional Experiments

**CIFAR-100.** We also validated our hypothesis with the ResNet20 trained on CIFAR-100. As shown in Fig. 6, the permuted solution consistently outperforms the naive solution, showing that our

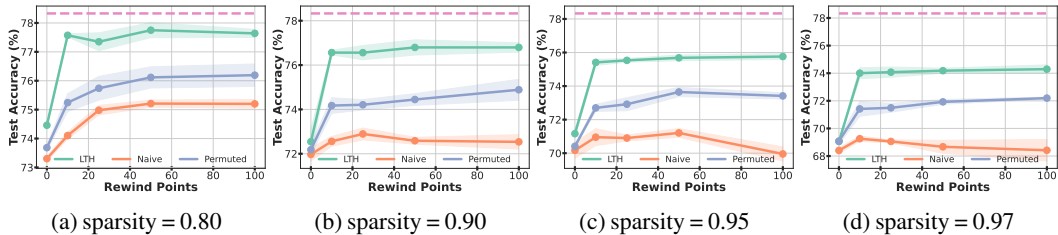

Figure 6: **ResNet20×{4}/CIFAR-100**. Results for the CIFAR-100 dataset with varying sparsities show that our permuted solution outperforms the naive solution. The dashed (- -) line shows the dense model accuracy.

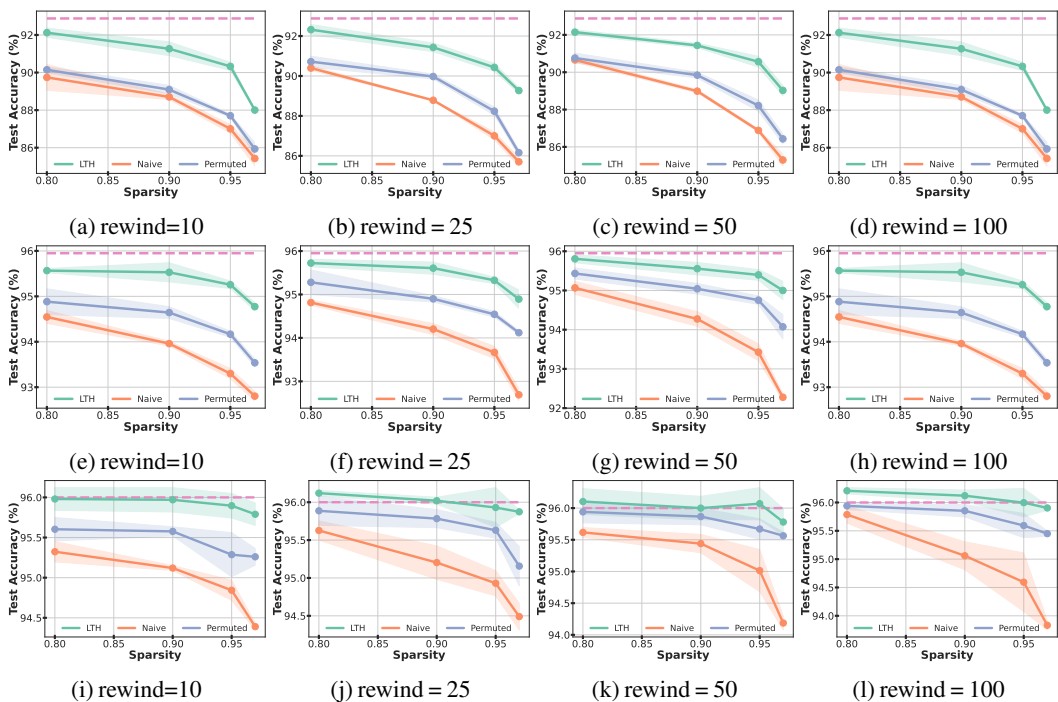

Figure 7: Accuracy vs sparsity trend for **ResNet20×{1,4,8}/CIFAR-10**. The top, middle, and bottom rows correspond to widths of $1$, $4$, and $8$, respectively. As the sparsity increases, the gap between permuted and naive solutions increases, showing permuted masks help with sparse training. With increased width and $k$, we observe a more significant gap seen throughout Figs. 7d, 7h and 7l and the permuted solution approaches the LTH solution. The dashed (- -) line shows the dense model accuracy.

hypothesis holds true across different models and datasets. Similar to the CIFAR-10 dataset, as we increase $k$, the performance of the permuted model improves.

### A.4  Additional Plots

Refer to Fig. 7 for additional accuracy vs sparsity plots across various widths.

### A.5  Experimental Methodology

The method outlined in Fig. 2, is also detailed fully below.

1. Randomly initialize two distinct neural networks from the Normal distribution: $\mathbf{w}_A^{t=0}, \mathbf{w}_B^{t=0} \sim \mathcal{N}$.

2. Train both networks to convergence for $T$ epochs: $\mathbf{w}_A^{t=T}, \mathbf{w}_B^{t=T}$.

3. Prune $\mathbf{w}_A^{t=T}$ via IMP (without weight-rewinding), producing a sparse subnetwork: $\mathbf{w}_A^{t=T} \odot \mathbf{m}_A$.

4. Perform activation matching by aligning the activations of $\mathbf{w}_A^{t=T}$ to $\mathbf{w}_B^{t=T}$, such that $\mathcal{B}(\pi(\mathbf{w}_A^{t=T}), \mathbf{w}_B^{t=T}) \leq \epsilon \implies$ error barrier below some threshold $\implies$ LMC.

5. Save checkpoints from step 2 at some epoch $t = k \ll T$, resulting in rewound epochs: $\mathbf{w}_A^{t=k}, \mathbf{w}_B^{t=k}$.

6. Sparse train the LTH solution: $\mathbf{w}_A^{t=k} \odot \mathbf{m}_A$ for $T-k$ epochs.

7. Sparse train the naive solution using the wrong initialization: $\mathbf{w}_B^{t=k} \odot \mathbf{m}_A$ for $T-k$ epochs.

8. Sparse train the permuted solution using the **permuted mask**: $\mathbf{w}_B^{t=k} \odot \pi(\mathbf{m}_A)$ for $T-k$ epochs.

## B  Time Complexity of the Permuted Solution

The primary difference in computational complexity between the LTH, naive, and permuted solutions lies in the process of neuronal alignment, where weight/activation matching is used to locate permutations in order to bring the hidden units of two networks into alignment. To obtain the permuted solution, two distinct models must be trained independently to convergence, after which their weights or activations are aligned through a permutation-matching process. This alignment, though relatively efficient, adds a small computational overhead compared to LTH and naive solutions, which do not involve matching steps. However, it's important to note that the primary goal of this study is not to improve training efficiency but rather to investigate why the LTH framework fails when applied to sparse training from new random initializations (not associated with the winning ticket's mask).

