# OpenReview forum: "Winning Tickets from Random Initialization: Aligning Masks for Sparse Training"
_NeurIPS.cc/2024/Workshop/UniReps — UniReps_

### Official Review · Reviewer_7zzp · 2024-10-06
**An innovative approach to mask alignment for sparse training, but practical efficiency remains questionable**

**Rating:** 5
**Confidence:** 4

**Review:**

This paper presents an interesting extension to the Lottery Ticket Hypothesis (LTH) by proposing a method to align sparse masks with new random initializations. The core idea is to use activation matching to permute a winning ticket mask, making it applicable to different initialization schemes, which addresses a key limitation of LTH—its sensitivity to initialization. While the proposed method demonstrates potential and novelty, its practical implications and efficiency need further justification. Here is a detailed evaluation of the work:

## Strengths:

1. **Novelty:** The paper introduces a creative solution to the limitation of LTH by leveraging weight symmetry and activation matching, allowing for the transfer of a sparse mask to a new random initialization. This approach is new and provides a fresh perspective on improving the generalizability of sparse models.

2. **Experimental Results:** The authors provide extensive experimental validation on CIFAR-10 and CIFAR-100 with ResNet and VGG architectures. The results suggest that the proposed permutation of masks improves generalization performance for sparse training with new initializations.

3. **Theoretical Motivation:** The discussion of weight symmetry and the alignment of optimization basins is well-motivated and grounded in existing literature. The proposed method offers a plausible hypothesis for why LTH masks fail to transfer across different random initializations.

## Weaknesses:

1. **Efficiency Concerns:** One of the major drawbacks of this approach is the need to train two models (A and B) and perform activation matching. This significantly increases the computational overhead. The question arises: if model B needs to be trained from scratch, why not apply pruning directly on model B rather than going through the permutation process? This diminishes the practical value of the proposed method.
2. **Lack of Comparison with Direct Pruning:** The paper does not compare the proposed method with a straightforward pruning approach on model B, which seems like a more natural solution given the need to retrain model B. A direct comparison would clarify whether the added complexity of mask permutation is worth the effort.
3. **Computational Complexity:** The paper does not delve into the computational cost of activation matching and mask permutation. Given the additional steps, it's critical to discuss how these affect the overall training time, particularly for wider or deeper models.

---

### Official Review · Reviewer_3FJ9 · 2024-10-06
**Hesitant as this is very interesting work, but the experiments are not fully convincing and the language could use work.**

**Rating:** 5
**Confidence:** 4

**Review:**

Summary:
Lottery Tickets (or matching subnetworks) are sparse subnetworks that can achieve the same performance as the dense model. However, the sparsity mask when using Iterative Magnitude Pruning does not generalize across random initializations, it appears to be unique to the weight initialization that the mask was generated on. This paper addresses this issue by connecting other weight permutation literature to find allow the other initializations to work better for a given sparsity mask. They do this by finding a weight permutation for the new initialization that matches the activations of the original weights. They show that permuting first can improve the performance of matching subnetworks with re-intialized weights.

Review:
This paper promotes a better understanding of lottery tickets, uncovering how permutation symmetries in the loss landscape ultimately affect these tickets. I think this is a very interesting question to tackle; however, I am not fully convinced by their results and some of their discussion points. I encourage the authors to continue this work because I believe this can be a great full paper if they address some of the concerns and fix their limitations. I do have reservations about accepting this paper, but believe this could provide insight and motivate follow-on work if accepted as an extended abstract. I ultimately leave this to the program chairs to set the bar for extended abstracts, so take my rating with a grain of salt. I will try my best to provide in depth comments.

I believe the paper slightly misunderstands the technical definition of “Lottery Tickets”, and the main results of [11], Linear 181 mode connectivity and the lottery ticket hypothesis. Lottery tickets are subnetworks that are trained from initialization that achieve similar generalization performance as the dense. The emphasis here is on “trained from initialization”. These are found using IMP and rewinding the weights back to initialization. The authors seem to exclude this from their definition of “rewinding”: “In practice however, subsequent work has shown that when training modestly-sized models requires using weight rewinding [11]”. As a reader who is very familiar with the LTH, I understood what they meant to say; however, readers who are not may learn incorrectly. The authors should fix this language in their paper before camera ready (if possible).

Furthermore, [11] shows that unstable networks do not have lottery tickets that can be found through standard IMP. This is because the trajectory of early training is highly sensitive to the order of the data. Instead, they show that IMP can only find “matching tickets” from a rewinding time k>0. Any subnetwork that is found by rewinding to a point early in training is disqualified from being a lottery ticket because it never sparsely trains from initialization. “Matching” is the key word in this paper. This technically doesn’t change your paper content too much; however, none of your results with k>0 validate the LTH, only some other “matching” hypothesis that you could argue or create instead.

The experiments with k=0 do, however, appropriately refer to the LTH. Though, these results are not as strong as the cases where k>0. The naive results is far too similar to both the LTH solution and your solution in k=0 for conclusive evidence. For k>0, these results are more conclusive, showing that weight permutation can fix some of the performance issues with reinitializing the network. The experiment results overall seem a bit weak for showing this. The naive performance is still relatively high; can this method actually fix settings where the naive reinitalization breaks? I would like to see this performance at significantly higher sparsities just before model collapse. I understand the authors will be adding Imagenet experiments & I assume more models in the next iteration of this work.

One of the limitations of this work is not fully due to the accuracy of activation matching; this work is fundamentally limited by the structure of the new random initializations. The authors claim: “We expect that given perfect permutation mapping, the permuted solution will match the LTH accuracy”.
I believe this does will not hold always hold with perfect permutation. For a mask generated on weight initialization A, this will only be guaranteed to hold for a new weight initialization B if B is a permutation of A. This is extremely unlikely. In other scenarios, B will be (at best) an approximate permutation of A, which can fail when B trains to a different minima than A. For example, let A=(0,1) and B=(10,0). There’s only two permutations here, the identity or flipping the two values. In both cases, B can possibly train to a different local minima and be trapped.

For future work, this paper should be addressing the fundamental questions better. For example, for what structure of B can you match the LTH accuracy? This may be more straight forward if you assume A finds a unique global minima.

I also believe the clarity of the writing can be fixed. I do not feel confident that readers who have not read the LTH closely would be able to understand this. For a few examples:

“any other random initialization sparsified using the winning ticket mask fails to achieve good generalization performance.”
Instead something like: “A winning ticket’s sparsity mask does not generalize to other weight initializations”

“Lottery Ticket Hypothesis (LTH) proposes to solve the sparse training problem by re-using the same initialization as used to train the pruned models”
Instead something like: “Frankle et al. propose to retroactively pick a sparsity mask for a given weight initialization by < explanation of IMP > “

“In practice however, when training even modestly-sized models, weight rewinding [11] is necessary”
This is incorrectly worded. Weight rewinding is always used. [11] rewinds to an early point in training, the original LTH paper rewinds to initialization.

I really like Figures 1 & 2. Figure 1 does an amazing job on illustrating the permutation symmetries of the loss landscape; however, there may be a few logical issues to fix. For a) you need to show the sparse initialization training back to the minima.

---

### Official Review · Reviewer_wCAu · 2024-10-06
**The submission proposes a method to improve the generalization performance of sparse training by permuting the winning ticket mask to align with the optimization basin of a new random initialization, leading to better performance in sparse training.**

**Rating:** 7
**Confidence:** 4

**Review:**

**Strengths:**
- The submission presents a clear and well-defined problem statement.
- The explanation of key concepts is strong, providing a solid understanding of the approach.
- The proposed method addresses the limitation of poor generalization when using the original winning ticket mask with different random initializations. By introducing mask permutation to align with the new optimization basin, the approach significantly enhances the effectiveness of sparse training without requiring the expensive iterative retraining process.

**Comments:**
- A clearer caption or explanation is needed for Figure 2 to enhance understanding.
- Algorithm 1 needs to be rewritten for greater clarity, emphasizing that step 7 represents the core contribution of the paper, while steps 6 and 8 serve as baselines.
- It would be beneficial to include a time analysis. The performance of the proposed method and two baselines are now not significantly different. A comparison of the computational cost between the two methods could provide valuable insights.

---

### Official Review · Reviewer_PbjU · 2024-10-06
**The observations regarding the limitations of the Lottery Ticket Network (LTN) in scenarios involving different random initializations are noteworthy, particularly the novel insights presented in the proposed method that leverage weight symmetry.**

**Rating:** 8
**Confidence:** 2

**Review:**

## Summary:
The Lottery Ticket Hypothesis (LTH) posits that a sparse "winning ticket" mask and its corresponding weights can achieve equivalent generalization performance to a dense model while utilizing fewer parameters. However, applying the winning mask to the weight with a different random initialization often fails to achieve good generalization performances. Recent research suggests that deep neural networks (DNNs) trained from various random initializations can locate solutions within the same optimization basin, accounting for weight symmetry. In this context, the author introduced a method to permute the winning ticket mask to align it with new random initializations, thereby enhancing the generalization performance of sparse training compared to the naive one with the un-permuted mask. The experimental results were conducted rigorously, demonstrating the efficacy and promising potential of the proposed method.

## Strength
- S1: The proposed method is well-motivated by the observations regarding weight symmetry. The challenges associated with different random initializations in the LTN need to be addressed.
- S2: Despite the extended abstract track, the experimental results were conducted rigorously and demonstrated significant potential.

## Weakness
- W1: The proposed method significantly relies on reference [1]. In lines 128 to 138, the authors also acknowledged that the activation matching algorithm operates to identify the local optimum solution through a greedy algorithm.

- W2: The proposed method still entails substantial computational overhead associated with weight rewinding. Weight rewinding can be regarded as tracking the lineage of weights throughout the training process, and this technique necessitates a sufficient number of rewind epochs to demonstrate its efficacy effectively. For instance, as indicated in Table 2, the appropriate value of \( k \) is at least 50, constituting a significant number of epochs in the experiments conducted with CIFAR-10.

While the weaknesses above are noted, they do not substantially detract from the authors' intriguing observations or the proposed method's novelty.

## Reference
- [1] Ainsworth, Samuel K., Jonathan Hayase, and Siddhartha Srinivasa. "Git re-basin: Merging models modulo permutation symmetries." arXiv preprint arXiv:2209.04836 (2022).

---

### Decision · Program_Chairs · 2024-10-10

**Decision:**

Accept

**Comment:**

In light of the positive reviewers' feedback and relevancy of the submission, we are pleased to accept this paper for presentation at UniReps 2024. We kindly ask the authors to incorporate the reviewers' suggestions and feedback in the final camera-ready version of the manuscript.